# Sequential Organ Failure Assessment Outperforms Quantitative Chest CT Imaging Parameters for Mortality Prediction in COVID-19 ARDS

**DOI:** 10.3390/diagnostics12010010

**Published:** 2021-12-22

**Authors:** Daniel Puhr-Westerheide, Jakob Reich, Bastian O. Sabel, Wolfgang G. Kunz, Matthias P. Fabritius, Paul Reidler, Johannes Rübenthaler, Michael Ingrisch, Dietmar Wassilowsky, Michael Irlbeck, Jens Ricke, Eva Gresser

**Affiliations:** 1Department of Radiology, University Hospital, LMU Munich, 81377 Munich, Germany; jakob.reich@med.uni-muenchen.de (J.R.); bastian.sabel@med.uni-muenchen.de (B.O.S.); wolfgang.kunz@med.uni-muenchen.de (W.G.K.); matthias.fabritius@med.uni-muenchen.de (M.P.F.); paul.reidler@med.uni-muenchen.de (P.R.); johannes.ruebenthaler@med.uni-muenchen.de (J.R.); michael.ingrisch@med.uni-muenchen.de (M.I.); jens.ricke@med.uni-muenchen.de (J.R.); eva.gresser@med.uni-muenchen.de (E.G.); 2Department of Anesthesiology, University Hospital, LMU Munich, 81377 Munich, Germany; dietmar.wassilowsky@med.uni-muenchen.de (D.W.); michael.irlbeck@med.uni-muenchen.de (M.I.)

**Keywords:** COVID-19, acute respiratory distress syndrome, artificial intelligence, pulmonary hypertension, computed tomography scan, multi-organ failure

## Abstract

(1) Background: Respiratory insufficiency with acute respiratory distress syndrome (ARDS) and multi-organ dysfunction leads to high mortality in COVID-19 patients. In times of limited intensive care unit (ICU) resources, chest CTs became an important tool for the assessment of lung involvement and for patient triage despite uncertainties about the predictive diagnostic value. This study evaluated chest CT-based imaging parameters for their potential to predict in-hospital mortality compared to clinical scores. (2) Methods: 89 COVID-19 ICU ARDS patients requiring mechanical ventilation or continuous positive airway pressure mask ventilation were included in this single center retrospective study. AI-based lung injury assessment and measurements indicating pulmonary hypertension (PA-to-AA ratio) on admission CT, oxygenation indices, lung compliance and sequential organ failure assessment (SOFA) scores on ICU admission were assessed for their diagnostic performance to predict in-hospital mortality. (3) Results: CT severity scores and PA-to-AA ratios were not significantly associated with in-hospital mortality, whereas the SOFA score showed a significant association (*p* < 0.001). In ROC analysis, the SOFA score resulted in an area under the curve (AUC) for in-hospital mortality of 0.74 (95%-CI 0.63–0.85), whereas CT severity scores (0.53, 95%-CI 0.40–0.67) and PA-to-AA ratios (0.46, 95%-CI 0.34–0.58) did not yield sufficient AUCs. These results were consistent for the subgroup of more critically ill patients with moderate and severe ARDS on admission (oxygenation index <200, n = 53) with an AUC for SOFA score of 0.77 (95%-CI 0.64–0.89), compared to 0.55 (95%-CI 0.39–0.72) for CT severity scores and 0.51 (95%-CI 0.35–0.67) for PA-to-AA ratios. (4) Conclusions: Severe COVID-19 disease is not limited to lung (vessel) injury but leads to a multi-organ involvement. The findings of this study suggest that risk stratification should not solely be based on chest CT parameters but needs to include multi-organ failure assessment for COVID-19 ICU ARDS patients for optimized future patient management and resource allocation.

## 1. Introduction

In the ongoing coronavirus disease 2019 (COVID-19) pandemic, the severe acute respiratory syndrome coronavirus 2 (SARS-CoV-2) has globally caused an enormous socio-economic burden and large numbers of deaths [1]. COVID-19 disease climbed to being the third leading cause of death in the US in 2020, according to the Centers for Disease Control [2]. Despite vaccination efforts, mutations of the virus have since led to ongoing infection waves affecting the course of human lives [3].

Many risk factors for severe and critical disease progression and for high mortality have been discussed, including age, male gender, high body mass index (BMI) and underlying comorbidities [4,5,6]. About 14% of laboratory-confirmed COVID-19 patients develop severe—and up to 5%, critical—courses of the disease, and mortality rates were found to be as high as 50% for critical cases [7]. Around 15–30% of COVID-19 inpatients require intensive care treatment and invasive ventilation, and a substantial subpopulation of around three quarters of intensive care unit (ICU) patients develop respiratory failure such as acute respiratory distress syndrome (ARDS) [8,9,10,11,12,13,14]. Pulmonary involvement in COVID-19 disease with the development of respiratory insufficiency and ARDS is considered as one of the major complications and drivers for disease progression to critical stages and for fatalities [15,16]. Therefore, chest CT scans have been regarded as a valuable tool for diagnosis and allocation purposes from the beginning of the pandemic [17]. In earlier studies, the extent of chest CT imaging findings was shown to be indicative for the severity of the disease, for admission to ICU, for invasive ventilation requirements, and for ECMO requirements [18,19,20,21,22,23,24,25]. Whereas the disease course remains incompletely understood, severe COVID-19 seems to be characterized by heterogeneous clinical features and multi-organ damage. Apart from ARDS, disease prognosis has largely been influenced by multi-organ involvement such as heart failure, kidney failure and liver damage [26,27].

The ongoing COVID-19 pandemic resulted in recurring shortages in ICU resources. Early risk stratification supports reasonable resource allocation in critical scenarios and fast decision making with regards to appropriate patient treatment. However, the predictive potential of different clinical and imaging findings is incompletely understood. The aim of this study was to evaluate the potential of AI-based quantitative chest CT severity scoring and of the pulmonary artery (PA) to ascending aorta (AA) ratio as an indicator for pulmonary hypertension in COVID-19 ICU patients requiring mechanical ventilation or CPAP mask ventilation for mortality prediction. Imaging biomarkers are compared to the performance of clinical scores on ICU admission, which reflect lung and multi-organ involvement in COVID-19 disease.

## 2. Materials and Methods

### 2.1. Patient Data

This retrospective single-center study was approved by the local institutional review board and included all COVID-19 patients (n = 89) requiring mechanical ventilation or non-invasive ventilation with continuous positive airway pressure (CPAP-mask), admitted to the participating ICUs from 03/2020 until 01/2021 and already discharged or deceased by end of January 2021, with positive SARS-CoV-2 PCR testing and CT scans within 48 h of hospital admission, as shown in the study flow chart (Figure 1).

Patient data were collected retrospectively and obtained from a digital patient information system (QCare PDMS, Health Information Management GmbH, Bad Homburg, Germany), which is routinely used in the corresponding ICUs. Collected data include age, gender, BMI, length of stay on ICU, in-hospital mortality, discharge from ICU, hours of invasive and non-invasive ventilation with continuous positive airway pressure (CPAP-mask), SOFA score, respiratory data as oxygenation indices, lung compliances and PEEP values. Chest CTs were obtained from our digital radiologic information system (RIS) and Picture Archiving and Communication System (PACS).

### 2.2. Image Acquisition

CT scans (n = 91) were performed using CT scanners of our emergency department (Siemens Somatom Force and Somatom AS+, Siemens Healthineers, Forchheim, Germany; GE Optima 660, GE Healthcare, Chicago, IL, USA), either as non-contrast high-resolution scan or with contrast-enhanced pulmonary embolism protocol at end inspiration with the patient in supine position. Image acquisition was modulated between 80 and 120 kVp with adaptive tube current (mAS). All images were reconstructed with a slice thickness of 1.00 mm or 1.25 mm. Multiplanar reconstruction methods were performed on all images. CT scans for n = 4 patients were performed at external hospitals before transfer to our hospital for ICU therapy with comparable scanning parameters. The datasets were suitable for AI-assessment and included in the present study.

### 2.3. Artificial-Intelligence-Based Quantification of Lung Involvement

A CE-certified CAD4COVID CT software tool (Thirona B.V., Nijmegen, The Netherlands) was used for the quantification of lung involvement on CT under the supervision of two radiologists with 4 and 7 years of clinical experience. CAD4COVID provides segmentation of lung lobes and demonstrates lung injury through a colored heatmap. The affected lung volume is quantified as percentage of the total lung volume (0–100%), and a score is generated ranging between 0 and 25, which indicates the extent of COVID-19 related abnormalities on the CT scan (0–5 points per lobe, maximum score 25 overall, Figure 2a). The performance of the CAD4COVID method in the detection of COVID-19 lung changes was rated as comparable with that of human readers, as shown in an evaluation study (26). CAD4COVID is a freely usable tool (CE-certified, class II, CE 0344), and access can be requested via the Thirona website (https://thirona.eu/cad4covid/, accessed on 2 December 2021). It is made available free-of-charge to support healthcare facilities during the pandemic. Axial lung kernel CT scans can be uploaded in DICOM file format after anonymization.

Further, the diameter ratio of the pulmonary artery (PA) to the ascending aorta (AA) was calculated as a measure for pulmonary hypertension. For this purpose, the diameter of the pulmonary trunk close to the bifurcation into the left and right main pulmonary artery and the ascending aorta were manually measured on the same frame, as shown in Figure 2b,c, and the ratio was calculated as a measure of potential pulmonary hypertension [28,29].

### 2.4. Prediction Parameters for the Regression Analysis

The demographic characteristics age, sex and BMI have been shown to be significant risk factors for disease severity and were therefore included in our regression model [12,23]. The SOFA score, which is based on representative scores for respiratory, cardiovascular, hepatic, coagulation, renal and neurological system function, and which rates these six different organ systems on a scale from zero to four points (range 0 to 24 points), was added as a measure for multi organ failure. Additionally, the severity score of the CAD4COVID tool was used as imaging feature for the quantification of lung involvement, and the PA-to-AA ratio was included as an indicator for pulmonary hypertension in COVID-19 patients [28,29,30].

### 2.5. Statistical Analysis

All statistical analyses were performed with SPSS software (version 26.0, IBM, Armonk, NY, USA). Continuous variables are reported as median with interquartile ranges (IQR). Mann–Whitney-U for continuous variables and Chi-square test or Fisher’s exact tests for categorical variables were applied to test for differences between survivors and non-survivors. Significance was defined as a two-sided *p*-value < 0.05. Binary logistic regression for the prediction of in-hospital mortality was performed adjusting for multiple covariates. Odds Ratios with 95% confidence intervals are shown. Receiver operating characteristic (ROC) analyses using exact binomial confidence intervals (CI) were used to compare the predictive performance of parameters, and the area under the curve (AUC) was calculated.

## 3. Results

### 3.1. Baseline Clinical Characteristics and Demographic Data

Median age of included patients was 65 years (IQR 53–73); 79% of patients were male, and median BMI was 27 (IQR 25–33). Median SOFA score on admission was 8 (IQR 6–11), median lactate on admission was 1.3 (IQR 1.0–1.8) and the median oxygenation index on admission was 168 (110–226), while 34% of patients had diabetes, 61% had hypertension, 34% had heart disease, 18% had pulmonary disease, 10% had chronic kidney disease, 10% had active malignancy and 8% had immunosuppression. Three patients (4%) had no ARDS on ICU admission, 24 (30%) had mild, 39 (48%) had moderate and 15 (19%) had severe. Median severity score on admission CT was 15 (IQR 11–20), median percentage of lung involvement on CT was 36 (IQR 20–57) and median pulmonary artery to ascending aorta (PA-to-AA) ratio was 0.86 (IQR 0.78–0.94). All baseline characteristics are shown in Table 1.

### 3.2. Differences in Clinical and Imaging Parameters for Survivors vs. Non-Survivors

The median age was significantly lower in surviving patients (n = 53, median age 62 years, IQR 52–70) compared to non-surviving patients (n = 36, median age 68 years, IQR 59–81), *p* = 0.005. ARDS severity on admission was not significantly different between the groups. The number of patients with mechanical ventilation and the hours on the ventilator were significantly higher in non-surviving patients (92% of patients vs. 72%, *p* = 0.021 and 340 h vs. 163 h, *p* = 0.038), whereas the number of patients on NIV (non-invasive ventilation with CPAP-mask) and the hours on NIV-ventilation were significantly higher in surviving patients (66% vs. 42%, *p* = 0.023 and 3 h vs. 0 h, *p* = 0.044). In the non-survivor group, significantly more patients were on hemodiafiltration (72% vs. 25%, <0.001) and significantly more patients had ECMO therapy (33% vs. 4%, *p* < 0.001). The SOFA score on admission, the maximum SOFA score and the mean SOFA score during ICU stay were significantly higher in the non-survivor group (*p* < 0.001 for all three parameters). There were significantly more patients with diabetes (47% vs. 25%, *p* = 0.026) and heart disease (53 vs. 21%, *p* = 0.002) in the non-survivor group. All parameters are shown in Appendix A.

The SOFA score on admission showed significant differences between survivors and non-survivors (7, IQR 4–10 vs. 11, IQR 8–14, *p* < 0.001), as shown in Figure 3a. The CT severity score on admission and the PA-to-AA ratio were not significantly different between the groups, and the oxygenation index on admission was not significantly different between the groups either. Lung compliance on admission was significantly lower in the non-survivor group, *p* = 0.011. All results are shown in Appendix A.

In addition, in the subgroup of patients with moderate or severe ARDS on ICU admission (n = 54), the SOFA score on admission showed significant differences between survivors and non-survivors (8, IQR 5–11 vs. 11, IQR 9–14, *p* = 0.023), as shown in Figure 3b. The CT severity score and the PA-to-AA ratio were not significantly different between survivors and non-survivors in the subgroup of moderate or severe ARDS on ICU admission.

### 3.3. Risk Stratification for In-Hospital Mortality

A multivariate binary logistic regression model was used to test for associations of demographic, clinical and imaging parameters on admission with in-hospital mortality. In the analysis including age, sex, BMI, SOFA on admission and the imaging parameters CT severity score on admission and PA-to-AA ratio, only age and the SOFA score on admission were significantly associated with in-hospital mortality (odds ratio for age 1.07 (95%-CI 1.00–1.13), *p* = 0.036 and for SOFA score on admission 1.05, (95%-CI 1.17–1.70), *p* < 0.001); results are shown in Table 2.

For the prediction of in-hospital mortality, an area under the curve (AUC) of 0.74 (95%-CI 0.63–0.85) for SOFA on admission and an AUC of 0.65 (95%-CI 0.53–0.77) for age was found in receiver operating characteristic (ROC) curves, as shown in Figure 4 and Table 3. ROC analyses for CT severity score on admission and for PA-to-AA ratio did not yield sufficient AUCs for mortality prediction (0.53 (95%-CI 0.40–0.67) and 0.46 (95%-CI 0.34–0.58), respectively). Other lung parameters such as the oxygenation index on admission, which is part of the SOFA score for the assessment of respiratory function, or the lung compliance on admission yielded AUCs inferior to SOFA score on admission (for oxygenation index on admission 0.63 (95%-CI 0.25–0.50), for lung compliance on admission 0.67 (95%-CI 0.55–0.79), Appendix A).

Additionally, ROC analysis for mortality prediction was performed in the subgroup of patients with moderate or severe ARDS on admission (oxygenation index <200) to evaluate if quantification of lung injury on chest CT has a higher predictive potential in patients with a lung-predominant disease manifestation. In those cases, the SOFA score on admission yielded an AUC of 0.77 (95%-CI 0.64–0.89), whereas age only yielded an AUC of 0.60 (95%-CI 0.44–0.75), as shown in Figure 5 and Table 4. Also in this subgroup, the CT severity score on admission and the PA-to-AA ratio did not yield sufficient AUCs for mortality prediction (0.55 (95%-CI 0.39–0.72) and (0.51 (95%-CI 0.35–0.67), respectively).

## 4. Discussion

Respiratory insufficiency and the development of an ARDS is seen as a main complication associated with high mortality rates in COVID-19 disease [15,16]. Risk stratification is often based on the extent of pulmonary involvement, and therefore chest CTs have played an important role in the triage of patients from the beginning of the pandemic [17,18,19,20]. However, it has been shown that severe COVID-19 disease is not limited to lung (vessel) injury but leads to a multi-organ involvement [26,31]. The aim of the study was to analyze the discriminative value of quantitative CT imaging biomarkers as well as of the SOFA score upon ICU admission of severely ill SARS-CoV-2 PCR-positive ICU patients (n = 89) who developed ARDS in need of invasive-ventilation therapy or CPAP-mask ventilation for mortality prediction.

AI-based quantification of lung involvement as well as PA-to-AA ratio as a measure for pulmonary hypertension in COVID-19 ICU patients on-admission CTs was not sufficient in predicting mortality in these cases (AUC = 0.53 (95%-CI 0.40–0.67) and 0.46 (95%-CI 0.34–0.58), respectively), which was also true even for the subgroup of moderate to severe ARDS patients (0.55 (95%-CI 0.39–0.72) and 0.45 (95%-CI 0.32–0.58), respectively). The paO2/FiO2 ratio and the lung compliance on admission as measures for lung dysfunction showed an inferior diagnostic value for mortality prediction in COVID-19 ARDS (Appendix A) compared to the SOFA score on admission. The SOFA score on admission is calculated with multiple clinical parameters that assess multi-organ dysfunction including the paO2/FiO2 ratio for oxygenation impairment. This score showed a robust predictive potential with an AUC of 0.74 (95%-CI 0.63–0.85) for the overall group (Table 4, Figure 4) and an AUC of 0.77 (95%-CI 0.64–0.89) in the moderate to severe ARDS group (paO2/FiO2 < 200 mmHg), (Table 4, Figure 5). In line with previous studies, age was found to be significantly associated with a fatal outcome in the multiparametric regression analysis but showed weak predictive power (overall study population with AUC 0.65 (95%-CI 0.53–0.77) and moderate to severe ARDS with AUC 0.60 (95%-CI 0.44–0.75) [12].

It has been shown that lung manifestations of COVID-19 disease on CT imaging are decisive for the evaluation of ARDS development or short-term progression to severe pneumonia as well as for assessment of invasive ventilation requirements [18,22,24]. When it comes to severe oxygenation impairment in ARDS, AI-derived severity scoring from CT showed substantial stratification potential at an early stage of the COVID-19 disease in predicting requirement of ECMO therapy, but the SOFA score on admission could even improve predictive capability of the AI-based CT model [25]. In this study, CT-based imaging features (severity scoring and PA-to-AA ratio) were not helpful for in-hospital mortality prediction and other parameters, indicating lung dysfunction (paO2/FiO2 ratio and lung compliance) was inferior for the prediction of fatal outcome in COVID-19 ARDS to the multiparametric SOFA score assessing multiple organ functions of patients. These findings suggest that lung involvement by itself is an important factor in explaining mortality risk in COVID-19 ARDS; however, it can be significantly improved by involving scores for multi-organ involvement. Several previous studies have evaluated the role of the SOFA score in the context of mortality in COVID-19, with inconsistent results. Whereas its usefulness as a risk stratification parameter was challenged in some studies [32], other studies showed a significant association of the SOFA score with mortality in COVID-19 [5,33,34] and with pneumonia severity in general [35]. The findings of this particular study support the assumption that the extent of multi-organ involvement in the COVID-19 disease and the extent of early multiple organ failure are decisive for the assessment of mortality risk. Patients developing ARDS from COVID-19 disease should not be triaged based on lung dysfunction alone. Clinical scores such as the SOFA score, reflecting multiple organ functions, seem to assess fatal disease courses more comprehensively and are therefore relevant tools for intensive care physicians to assess patients’ risk for mortality as early as upon ICU admission.

### Limitations

The single-center retrospective analysis with a limited sample size due to ICU capacity restrictions for the treatment of severe COVID-19 patients in our hospital is a limitation and requires further, ideally larger external validation cohorts. Secondly, discharged patients were not followed up after dismissal from our hospital.

## 5. Conclusions

CT imaging features of pulmonary changes on admission CTs proved insufficient for mortality prediction in COVID-19 ARDS patients. Clinical parameters for lung dysfunction such as the oxygenation index and lung compliance partially reflect overall disease severity but the more comprehensive clinical SOFA score for multi-organ assessment outperformed lung-centered parameters for mortality prediction. Therefore, the SOFA score allows for an optimized triage upon ICU admission of COVID-19 ARDS patients. These findings should be validated in future prospective multicenter studies.

## Figures and Tables

**Figure 1 diagnostics-12-00010-f001:**
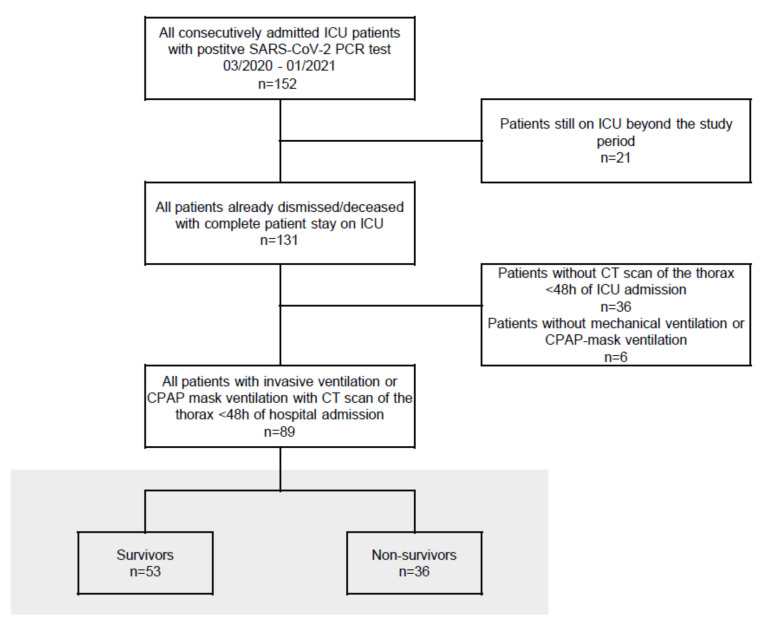
Study flow chart. All patients from the participating ICUs with positive SARS-CoV PCR test between 2 March 2020 and 26 January 2021, who were discharged or deceased, required mechanical ventilation or CPAP-mask ventilation and received a CT scan of the thorax on admission, were included.

**Figure 2 diagnostics-12-00010-f002:**
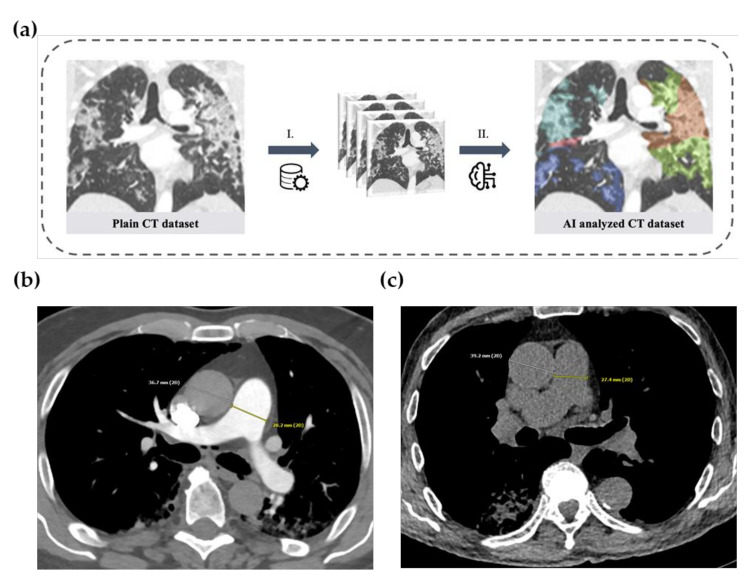
(**a**). CT scans of the thorax assessed with artificial intelligence: lung injury was quantified using a severity score ranging from 0 to 25. Left: Plain coronal slice image of the thorax CT scan in a patient with SARS-CoV-2 positive PCR test. Right: AI analyzed CT dataset with color staining of affected lung tissue (colored areas in each lung lobe). I. Pre-processing with data anonymization and uploading. II. AI-based analysis of CT dataset. (**b**,**c**). Measurements of the diameter of the ascending aorta (AA) and the pulmonary trunk (PA) close to the bifurcation in enhanced (**b**) and unenhanced (**c**) CT images for the calculation of the PA-to-AA ratio.

**Figure 3 diagnostics-12-00010-f003:**
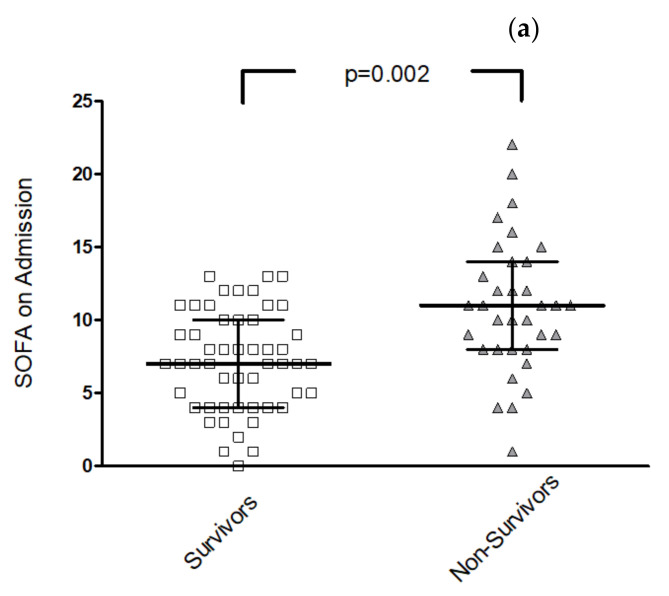
(**a**) SOFA score for all included patients. Left: survivors (squares), right: non-survivors (triangles). Lines show median, bars show interquartile ranges. (**b**). SOFA score for patients with ARDS Type 2 or 3 (moderate or severe, oxygenation index < 200). Left: survivors, right: non-survivors.

**Figure 4 diagnostics-12-00010-f004:**
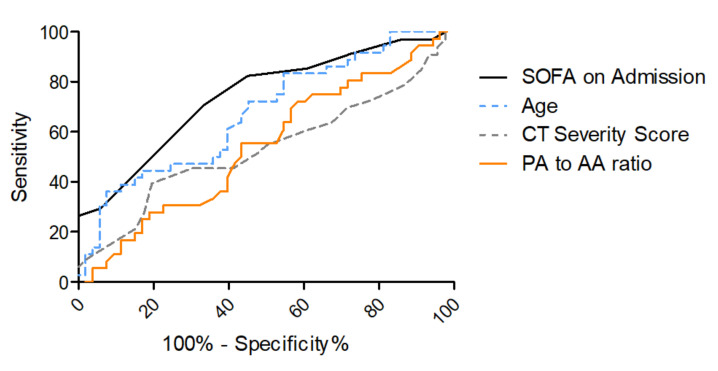
ROC curves for SOFA score on ICU admission and age for the prediction of in-hospital mortality. ROC curve for in-hospital mortality according to SOFA score on admission (grey), AUC = 0.74 (95% CI 0.63–0.85), age (dashed blue), AUC = 0.65 (95% CI 0.53–0.77), CT severity score (dashed grey), AUC = 0.53 (95%-CI 0.40–0.67) and PA-to-AA ratio (orange), AUC = 0.45 (95%-CI 0.32–0.58).

**Figure 5 diagnostics-12-00010-f005:**
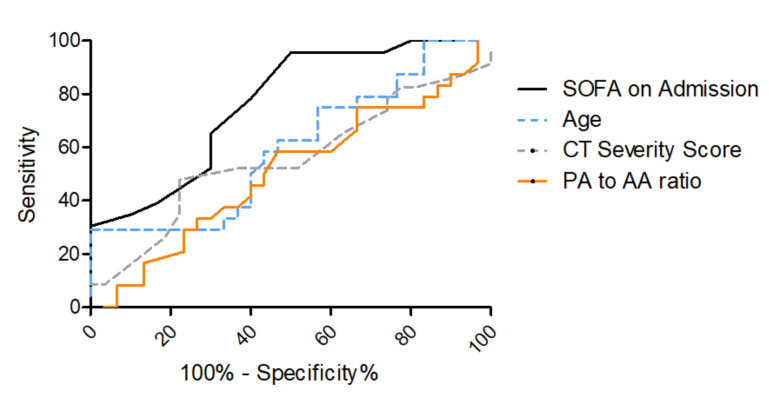
ROC curves for SOFA score on ICU admission and age for the prediction of in-hospital mortality in patients with moderate or severe ARDS on ICU admission. ROC curve for in-hospital mortality according to SOFA score on admission (grey), AUC = 0.77 (95% CI 0.64–0.89) and age (dashed blue), AUC = 0.65 (95% CI 0.42–0.74).

**Table 1 diagnostics-12-00010-t001:** Baseline characteristics of COVID-19 study patients admitted to the participating ICUs. Values presented are count (percentage) for categorical and median (interquartile range) for ordinal or continuous variables. COVID-19, Coronavirus disease 2019; ICU, intensive care unit; SOFA score, sequential organ failure assessment score; ARDS, acute respiratory distress syndrome; and CT, computed tomography; * 4 missing values; ** 9 missing values. *** 8 missing values. **** 10 missing values.

	COVID-19 ICU-Patients (n = 89)		
**Patient Data**			
Age	65		(53–73)
Male Sex	70		(78.7%)
Body Mass Index	27		(25–33)
SOFA Score on Admission *	8		(6–11)
Lactate on Admission	1.3		(1.0–1.8)
Oxygenation Index on Admission **	168		(110–226)
**Comorbidities**			
Diabetes		31	(34.4%)
Hypertension		55	(61.1%)
Heart Disease		31	(34.4%)
Pulmonary Disease		16	(17.8%)
Chronic Kidney Disease		9	(10.1%)
Active Malignancy		9	(10.1%)
Immunosuppression		7	(7.8%)
**ARDS Type on Admission *****			
Mild	24		(29.6%)
Moderate	39		(48.1%)
Severe	15		(18.5%)
No ARDS on Admission	3		(3.7%)
**CT Features on Admission**			
CT-Severity Score ****	15		(11–20)
CT-Percentage of Lung Involvement ****	36		(20–57)
Pulmonary artery to ascending aorta ratio	0.86		(0.78–0.94)

**Table 2 diagnostics-12-00010-t002:** Predictors for In-Hospital Mortality for COVID-19 ICU patients (n = 75). Results from binary logistic regression analysis with adjustment for multiple covariates. COVID-19, coronavirus disease 2019; CI, confidence interval; ICU, intensive care unit; BMI, body mass index; SOFA, sequential organ failure assessment; CT, computed tomography; PA, pulmonary artery; AA, ascending aorta; and * statistically significant (*p* < 0.05).

	In-Hospital-Mortality		
Independent Variables	Odds Ratio	CI	*p* Value
Age	1.067	1.004–1.134	0.036 *
Sex	0.231	0.048–1.118	0.069
BMI	1.044	0.936–1.164	0.444
SOFA on Admission	1.409	1.171–1.696	<0.001 *
CT Severity Score on Admission	1.046	0.941–1.163	0.402
PA-to-AA Ratio	0.086	0.001–12.934	0.337

**Table 3 diagnostics-12-00010-t003:** ROC analysis for the prediction of mortality with Youden index, sensitivity and specificity. AUC, area under the curve; CI, confidence interval; Y-index; Youden index; and SOFA, sequential organ failure assessment score.

N = 53 Survivors, N = 36 Non-Survivors						
Survivors (n = 51) vs. Non-Survivors (n = 34)	AUC (95% CI)		Y-Index	Discriminative Value	Sensitivity	Specificity
**SOFA Score on Admission**	0.74	0.63–0.85	0.37	7.5	0.82	0.55
Survivors (n = 53) vs. Non-Survivors (n = 36)						
**Age**	0.68	0.56–0.79	0.26	57.7	0.83	0.45

**Table 4 diagnostics-12-00010-t004:** ROC analysis for the prediction of mortality with Youden index, sensitivity and specificity. AUC, area under the curve; CI, confidence interval; Y-index; Youden index; and SOFA, sequential organ failure assessment score.

N = 30 Survivors, N = 23 Non-Survivors						
Survivors (n = 30) vs. Non-Survivors (n = 23)	AUC (95% CI)		Y-Index	Discriminative Value	Sensitivity	Specificity
**SOFA Score on Admission**	0.77	0.64–0.89	0.46	7.5	0.96	0.50
Survivors (n = 30) vs. Non-Survivors (n = 24)						
**Age**	0.60	0.44–0.75	0.29	57.4	0.29	1.00

## Data Availability

The datasets analyzed during the current study are available from the corresponding author on reasonable request. The CAD4COVID tool, which was used to analyze the CT data sets, is a CE-certified tool that was made available freely by Thirona B.V., Nijmegen, Netherlands (URL https://thirona.eu/cad4covid/, accessed on 3 June 2021).

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
