# Peer review of "Sequential Organ Failure Assessment Outperforms Quantitative Chest CT Imaging Parameters for Mortality Prediction in COVID-19 ARDS"

_diagnostics, 2021, doi:10.3390/diagnostics12010010_

Round 1

Reviewer 1 Report

In this paper, the authors described Clinical Scores Outperform AI-based Severity Scoring and Pulmonary Hypertension Imaging Parameters on Chest CTs for Mortality Prediction in COVID-19 ARDS Patients. The topic of this research is relevant to the scope of the journal, and I believe the authors have executed an interesting study. In my opinion, the paper is suitable for publication, but a major  revision is needed. The following specific comments can also be considered to improve the quality of this paper:

  1. First of all, I believe the title of the paper is not appropriate. Although the scope of the paper is
    interesting, but the title should be shorter
  2. Your abstract should be shorter as well. 
  3. Please revise the abstract and highlight the contribution of your research in the abstract. In addition,
    it would be a good idea to add one sentence to your abstract to state your motivation.
  4. Problem definition is a bit unclear, please define your problem in a systematic way.
  5. The conclusion section should be improved, and directions for future studies should be stated.
  6. All the citations in the manuscript need to be double-checked to ensure its consistency with the
    reference list.
  7. The contribution of this study has to be described more clearly in the introduction section or a
    separate section. 
  8. Conclusion section needs substantial revisions to include the actual outcomes and practical implications.
  9. Please expalin the folloeing paragraph in page 8, It is unclear: Additionally, we performed ROC analysis for mortality prediction in the subgroup of patients with moderate or severe ARDS on admission (oxygenation index < 200). In these patients, the SOFA score on admission yielded an AUC of 0.77 (95%-CI 0.64-0.89), whereas age only yielded an AUC of 0.60 (95%-CI 0.44-0.75), as shown in Figure 5 and
    Table 5. Also in this subgroup, the CT severity score on admission and the PA to AA ratio did not yield sufficient AUCs for mortality prediction (0.55 (95%-CI 0.39-0.72) and (0.51 (95%-CI 0.35-0.67), respectively).
  10. Please do not use I and we in your manuscript(first person singular and plural).
  11. The paper's writing should be enhanced. A further proofreading by a professional English editor
    is needed.

Reviewer 2 Report

The study does not provide any new hint to the understanding of the development if the disease. 

The study confirms a multi-organ involvement in the COVID-19 disease.

The authors state that only age and the SOFA score on admission were significantly associated with in-hospital mortality. 

However, the authors also recognize  that "Lung compliance on admission was significantly lower in the non-survivor group". Accordingly, one cannot fully share the conclusion that "Clinical parameters for lung dysfunction such as....... lung compliance have a limited predictive potential and are outperformed by the clinical SOFA score".

Reviewer 3 Report

The manuscript is decent. But I have one serious question needed to be addressed before thinking for publication.

Is sample number 152 is good for AI-based studies?. How do u justify?

Round 2

Reviewer 1 Report

Accept in present form.

Reviewer 3 Report

The manuscript needs some improvement in the introduction, so the readers will be interested in reading it. It should have an excellent introduction that should link COVID with AI. 

Some of the recent manuscript should be cited below.

  1. https://www.nature.com/articles/s41467-020-17971-2
  2. https://www.frontiersin.org/articles/10.3389/frcmn.2021.645040/full
  3. https://onlinelibrary.wiley.com/doi/full/10.1002/mbo3.1122
  4. https://www.sciencedirect.com/science/article/pii/S2590137021000121